# ATTENTION FORCING FOR SEQUENCE-TO-SEQUENCE MODEL TRAINING

## ABSTRACT

Auto-regressive sequence-to-sequence models with attention mechanism have achieved state-of-the-art performance in many tasks such as machine translation and speech synthesis. These models can be difficult to train. The standard approach, teacher forcing, guides a model with reference output history during training. The problem is that the model is unlikely to recover from its mistakes during inference, where the reference output is replaced by generated output. Several approaches deal with this problem, largely by guiding the model with generated output history. To make training stable, these approaches often require a heuristic schedule or an auxiliary classifier. This paper introduces attention forcing, which guides the model with generated output history and reference attention. This approach can train the model to recover from its mistakes, in a stable fashion, without the need for a schedule or a classifier. In addition, it allows the model to generate output sequences aligned with the references, which can be important for cascaded systems like many speech synthesis systems. Experiments on speech synthesis show that attention forcing yields significant performance gain. Experiments on machine translation show that for tasks where various re-orderings of the output are valid, guiding the model with generated output history is challenging, while guiding the model with reference attention is beneficial.

## 1 INTRODUCTION

Auto-regressive sequence-to-sequence (seq2seq) models with attention mechanism are widely used in a variety of areas including Neural Machine Translation (NMT) (Neubig, 2017; Huang et al., 2016) and speech synthesis (Shen et al., 2018; Wang et al., 2018), also known as Text-To-Speech (TTS). These models excel at connecting sequences of different length, but can be difficult to train. A standard approach is teacher forcing, which guides a model with reference output history during training. This makes the model unlikely to recover from its mistakes during inference, where the reference output is replaced by generated output. One alternative is to train the model in free running mode, where the model is guided by generated output history. This approach often struggles to converge, especially for attention-based models, which need to infer the correct output and align it with the input at the same time.

Several approaches are introduced to tackle the above problem, namely scheduled sampling (Bengio et al., 2015) and professor forcing (Lamb et al., 2016). Scheduled sampling randomly decides, for each time step, whether the reference or generated output token is added to the output history. The probability of choosing the reference output token decays from 1 to 0 with a heuristic schedule. A natural extension is sequence-level scheduled sampling, where the decision is made for each sequence instead of token. Professor forcing views the seq2seq model as a generator. During training, the generator operates in both teacher forcing mode and free running mode. In teacher forcing mode, it tries to maximize the standard likelihood. In free running mode, it tries to fool a discriminator, which is trained to tell if the model is running in teacher forcing mode. To make training stable, the above approaches require either a well tuned schedule, or a well trained discriminator.

This paper introduces attention forcing, which guides the model with generated output history and reference attention. This approach makes training stable by decoupling the learning of the output and that of the alignment. There is no need for a schedule or a discriminator. Furthermore, for cascaded systems like many TTS systems, attention forcing can be particularly useful. A model

trained with attention forcing can generate (in attention forcing mode) output sequences aligned with the references. These output sequences can be used to train a downstream model, enabling it to fix some upstream errors. The TTS experiments show that attention forcing yields significant gain in speech quality. The NMT experiments show that for tasks where various re-orderings of the output are valid, guiding the model with generated output history can be problematic, while guiding the model with reference attention yields slight but consistent gain in BLEU score (Papineni et al., 2002).

## 2 SEQUENCE-TO-SEQUENCE GENERATION

Sequence-to-sequence generation can be defined as the problem of mapping an input sequence $\boldsymbol{x}_{1:L}$ to an output sequence $\boldsymbol{y}_{1:T}$. From a probabilistic perspective, a model $\boldsymbol{\theta}$ estimates the distribution of $\boldsymbol{y}_{1:T}$ given $\boldsymbol{x}_{1:L}$, typically as a product of distributions conditioned on output history:

$$p(\boldsymbol{y}_{1:T}|\boldsymbol{x}_{1:L};\boldsymbol{\theta}) = \prod_{t=1}^{T} p(\boldsymbol{y}_t|\boldsymbol{y}_{1:t-1},\boldsymbol{x}_{1:L};\boldsymbol{\theta}) \tag{1}$$

Ideally, the model is trained through minimizing the KL-divergence between the true distribution $p(\boldsymbol{y}_{1:T}|\boldsymbol{x}_{1:L})$ and the estimated distribution:

$$\hat{\boldsymbol{\theta}} = \underset{\boldsymbol{\theta}}{\operatorname{argmin}} \, \mathbb{E}_{\boldsymbol{x}_{1:L}\sim p(\boldsymbol{x}_{1:L})} \mathrm{KL}\big(p(\boldsymbol{y}_{1:T}|\boldsymbol{x}_{1:L})||p(\boldsymbol{y}_{1:T}|\boldsymbol{x}_{1:L};\boldsymbol{\theta})\big)$$

$$= \underset{\boldsymbol{\theta}}{\operatorname{argmin}} \, \mathbb{E}_{\boldsymbol{x}_{1:L}\sim p(\boldsymbol{x}_{1:L})} \mathbb{E}_{\boldsymbol{y}_{1:T}\sim p(\boldsymbol{y}_{1:T}|\boldsymbol{x}_{1:L})} \log \big(p(\boldsymbol{y}_{1:T}|\boldsymbol{x}_{1:L})/p(\boldsymbol{y}_{1:T}|\boldsymbol{x}_{1:L};\boldsymbol{\theta})\big) \tag{2}$$

In practice, this is approximated by minimizing the Negative Log-Likelihood (NLL) of some training data $\{\boldsymbol{y}_{1:T}^{(n)}, \boldsymbol{x}_{1:L}^{(n)}\}_1^N$, sampled from the true distribution:

$$\hat{\boldsymbol{\theta}} = \underset{\boldsymbol{\theta}}{\operatorname{argmin}} - \sum_{n=1}^{N} \log p(\boldsymbol{y}_{1:T}^{(n)}|\boldsymbol{x}_{1:L}^{(n)};\boldsymbol{\theta}) \tag{3}$$

While $L$ and $T$ are functions of $n$, the subscripts are omitted to simplify notations, i.e. $L_n$ and $T_n$ are written as $L$ and $T$. At inference stage, given an input $\boldsymbol{x}_{1:L}^*$, the output $\hat{\boldsymbol{y}}_{1:T}$ can be obtained through searching for the most probable sequence from the estimated distribution:

$$\hat{\boldsymbol{y}}_{1:T} = \underset{\boldsymbol{y}_{1:T}}{\operatorname{argmax}} \, p(\boldsymbol{y}_{1:T}|\boldsymbol{x}_{1:L}^*;\hat{\boldsymbol{\theta}}) \tag{4}$$

The exact search is computationally expensive, and is often approximated by greedy search if the output space is continuous, or beam search if the output space is discrete (Bengio et al., 2015).

### 2.1 ATTENTION-BASED SEQ2SEQ MODEL

Attention mechanisms (Bahdanau et al., 2014; Chorowski et al., 2015) are commonly used to connect sequences of different length. This paper focuses on attention-based encoder-decoder models. For these models, the probability $p(\boldsymbol{y}_t|\boldsymbol{y}_{1:t-1},\boldsymbol{x}_{1:L};\boldsymbol{\theta})$ is estimated as:

$$p(\boldsymbol{y}_t|\boldsymbol{y}_{1:t-1},\boldsymbol{x}_{1:L};\boldsymbol{\theta}) \approx p(\boldsymbol{y}_t|\boldsymbol{y}_{1:t-1},\boldsymbol{\alpha}_t,\boldsymbol{x}_{1:L};\boldsymbol{\theta}) \approx p(\boldsymbol{y}_t|\boldsymbol{s}_t,\boldsymbol{c}_t;\boldsymbol{\theta}_y) \tag{5}$$

$$\boldsymbol{s}_t = f(\boldsymbol{y}_{1:t-1};\boldsymbol{\theta}_s) \tag{6}$$

$$\boldsymbol{c}_t = f(\boldsymbol{\alpha}_t,\boldsymbol{x}_{1:L};\boldsymbol{\theta}_c) \tag{7}$$

$\boldsymbol{\theta} = \{\boldsymbol{\theta}_y, \boldsymbol{\theta}_s, \boldsymbol{\theta}_c\}$. $\boldsymbol{\alpha}_t$ is an alignment vector (a set of attention weights). $\boldsymbol{s}_t$ is a state vector representing the output history $\boldsymbol{y}_{1:t-1}$, and $\boldsymbol{c}_t$ is a context vector summarizing $\boldsymbol{x}_{1:L}$ for the prediction of $\boldsymbol{y}_t$. The following equations, as well as figure 1, give a more detailed illustration of how $\boldsymbol{\alpha}_t$, $\boldsymbol{s}_t$ and $\boldsymbol{c}_t$ can be computed:

$$\boldsymbol{h}_{1:L} = f(\boldsymbol{x}_{1:L};\boldsymbol{\theta}_h) \tag{8}$$

$$\boldsymbol{s}_t = f(\boldsymbol{s}_{t-1},\boldsymbol{y}_{t-1};\boldsymbol{\theta}_s) \tag{9}$$

$$\boldsymbol{\alpha}_t = f(\boldsymbol{s}_t,\boldsymbol{h}_{1:L};\boldsymbol{\theta}_\alpha) \tag{10}$$

$$\boldsymbol{c}_t = \sum_{l=1}^{L} \alpha_{t,l}\boldsymbol{h}_l \tag{11}$$

$$\hat{\boldsymbol{y}}_t \sim p(\boldsymbol{y}_t|\boldsymbol{s}_t,\boldsymbol{c}_t;\boldsymbol{\theta}_y) \tag{12}$$

First the encoder maps $\boldsymbol{x}_{1:L}$ to an encoding sequence $\boldsymbol{h}_{1:L}$. For each decoder time step, $\boldsymbol{s}_t$ is updated with $\boldsymbol{y}_{t-1}$. Based on $\boldsymbol{h}_{1:L}$ and $\boldsymbol{s}_t$, the attention mechanism computes $\boldsymbol{\alpha}_t$, and then $\boldsymbol{c}_t$ as the weighted sum of $\boldsymbol{h}_{1:L}$. Finally, the decoder estimates a distribution based on $\boldsymbol{s}_t$ and $\boldsymbol{c}_t$, and optionally generates an output token $\hat{\boldsymbol{y}}_t$ by either sampling or taking the most probable token. Note that the output history $\boldsymbol{y}_{1:t-1}$ plays an important role, as it impacts $p(\boldsymbol{y}_t|\boldsymbol{s}_t,\boldsymbol{c}_t;\boldsymbol{\theta}_y)$ through both $\boldsymbol{s}_t$ and $\boldsymbol{c}_t$. Also note that there are many forms of attention-based encoder-decoder models. While attention forcing is illustrated with this particular form, it is not limited to it.

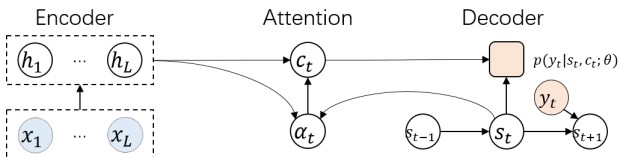

Figure 1: Illustration of an attention-based encoder-decoder model

## 2.2 TRAINING APPROACHES

As shown in equations 2 and 3, minimizing the KL-divergence between the true distribution and the model distribution can be approximated by minimizing the NLL. This motivates the approach to train the model in teacher forcing mode, where $p(\boldsymbol{y}_t|\boldsymbol{y}_{1:t-1}, \boldsymbol{x}_{1:L}; \boldsymbol{\theta})$ is computed with the correct output history $\boldsymbol{y}_{1:t-1}$, as shown in equations 5 and 6. In this case, the loss can be written as:

$$\mathcal{L}_y^{(\text{T})}(\boldsymbol{\theta}) = -\sum_{n=1}^{N} \log p(\boldsymbol{y}_{1:T}^{(n)}|\boldsymbol{x}_{1:L}^{(n)}; \boldsymbol{\theta}) = -\sum_{n=1}^{N} \sum_{t=1}^{T} \log p(\boldsymbol{y}_t^{(n)}|\boldsymbol{y}_{1:t-1}^{(n)}, \boldsymbol{x}_{1:L}^{(n)}; \boldsymbol{\theta}) \qquad (13)$$

This approach yields the correct model (zero KL-divergence) if the following assumptions hold: 1) the model is powerful enough ; 2) the model is optimized correctly; 3) there is enough training data to approximate the expectation shown in equation 2. In practice, these assumptions are often not true, hence the model is prone to make mistakes. To illustrate the problem, suppose there is a reference output $\boldsymbol{y}_{1:T}^*$ for the test input $\boldsymbol{x}_{1:L}^*$. Due to data sparsity in high-dimensional space, $\boldsymbol{x}_{1:L}^*$ is likely to be unseen during training. If the probability $p(\boldsymbol{y}_t^*|\boldsymbol{y}_{1:t-1}^*, \boldsymbol{x}_{1:L}^*; \boldsymbol{\theta})$ is wrongly estimated to be small at time step $t$, the probability of the reference output sequence $p(\boldsymbol{y}_{1:T}^*|\boldsymbol{x}_{1:L}^*; \boldsymbol{\theta})$ will also be small, i.e. it will be unlikely for the model to generate $\boldsymbol{y}_{1:T}^*$.

In practice, the model can be assessed by some loss $\mathcal{D}(\boldsymbol{y}_{1:T}^*, \hat{\boldsymbol{y}}_{1:T})$ between the reference output $\boldsymbol{y}_{1:T}^*$ and the generated output $\hat{\boldsymbol{y}}_{1:T}$. Taking the expected value yields the Bayes risk: $\mathbb{E}_{\hat{\boldsymbol{y}}_{1:T} \sim p(\boldsymbol{y}_{1:T}|\boldsymbol{x}_{1:L}^*; \boldsymbol{\theta})} \mathcal{D}(\boldsymbol{y}_{1:T}^*, \hat{\boldsymbol{y}}_{1:T})$. This motivates training the model with the following loss:

$$\begin{aligned} \mathcal{L}_y^{(\text{B})}(\boldsymbol{\theta}) &= \sum_{n=1}^{N} \mathbb{E}_{\hat{\boldsymbol{y}}_{1:T} \sim p(\boldsymbol{y}_{1:T}|\boldsymbol{x}_{1:L}^{(n)}; \boldsymbol{\theta})} \mathcal{D}(\boldsymbol{y}_{1:T}^{(n)}, \hat{\boldsymbol{y}}_{1:T}) \\ &\approx \sum_{n=1}^{N} \sum_{m=1}^{M} p(\hat{\boldsymbol{y}}_{1:T}^{(n,m)}|\boldsymbol{x}_{1:L}^{(n)}; \boldsymbol{\theta}) \mathcal{D}(\boldsymbol{y}_{1:T}^{(n)}, \hat{\boldsymbol{y}}_{1:T}^{(n,m)}) \end{aligned} \qquad (14)$$

$\hat{\boldsymbol{y}}^{(n,m)}$ is sampled from the estimated distribution $p(\boldsymbol{y}_{1:T}|\boldsymbol{x}_{1:L}^{(n)}; \boldsymbol{\theta})$. $\mathcal{D}$ is minimal when the two sequences are equal. So the model is trained to not only assign high probability to the reference sequences in the training data, but also assign low probability to other sequences. This makes minimum Bayes risk training prone to overfitting.

Very often, $\mathcal{D}$ is computed at sub-sequence level. Examples include BLEU score for NMT, word error rate for speech recognition and root mean square error for TTS. So if an approach trains the model to predict the reference output, based on erroneous output history, it will indirectly reduce the Bayes risk. One example is to train the model in free running mode, where $p(\boldsymbol{y}_t|\boldsymbol{y}_{1:t-1}, \boldsymbol{x}_{1:L}; \boldsymbol{\theta})$ is estimated with the generated output history:

$$p(\boldsymbol{y}_t|\boldsymbol{y}_{1:t-1}, \boldsymbol{x}_{1:L}; \boldsymbol{\theta}) \approx p(\boldsymbol{y}_t|\hat{\boldsymbol{y}}_{1:t-1}, \boldsymbol{x}_{1:L}; \boldsymbol{\theta}) \approx p(\boldsymbol{y}_t|\boldsymbol{s}_t, \boldsymbol{c}_t; \boldsymbol{\theta}_y) \qquad (15)$$

$$\boldsymbol{s}_t = f(\hat{\boldsymbol{y}}_{1:t-1}; \boldsymbol{\theta}_s) \qquad (16)$$

$\hat{\boldsymbol{y}}_t$ is obtained from the estimated distribution $p(\boldsymbol{y}_t|\boldsymbol{s}_t, \boldsymbol{c}_t; \boldsymbol{\theta}_y)$, as shown in equation 12. (The approaches discussed in this section are designed for all auto-regressive models, with or without attention mechanism. So the realization $\boldsymbol{c}_t$ is not shown.) The corresponding loss function is:

$$\mathcal{L}_y^{(\text{F})}(\boldsymbol{\theta}) = -\sum_{n=1}^{N} \sum_{t=1}^{T} \log p(\boldsymbol{y}_t^{(n)}|\hat{\boldsymbol{y}}_{1:t-1}^{(n)}, \boldsymbol{x}_{1:L}^{(n)}; \boldsymbol{\theta}) \qquad (17)$$

Note that if there is enough data and modeling power, and the model is optimized correctly, the distribution $\prod_{t=1}^{T} p(\boldsymbol{y}_t|\hat{\boldsymbol{y}}_{1:t-1}, \boldsymbol{x}_{1:L}; \boldsymbol{\theta})$ can be the same as the true distribution $p(\boldsymbol{y}_{1:T}|\boldsymbol{x}_{1:L})$. The problem with this approach is that training often struggles to converge. One concern is that the model needs to learn to infer the correct output and align that with the input at the same time. Therefore, several approaches, namely scheduled sampling and professor forcing, are proposed to train the model in a mode between teacher forcing and free running.

Scheduled sampling (Bengio et al., 2015) randomly decides, for each time step, whether the reference or generated output token is added to the output history $\widetilde{\boldsymbol{y}}_{1:t-1}$. For this approach,

$p(\boldsymbol{y}_t|\boldsymbol{y}_{1:t-1}, \boldsymbol{x}_{1:L}; \boldsymbol{\theta})$ is estimated as:

$$p(\boldsymbol{y}_t|\boldsymbol{y}_{1:t-1}, \boldsymbol{x}_{1:L}; \boldsymbol{\theta}) \approx p(\boldsymbol{y}_t|\widetilde{\boldsymbol{y}}_{1:t-1}, \boldsymbol{x}_{1:L}; \boldsymbol{\theta}) \approx p(\boldsymbol{y}_t|\boldsymbol{s}_t, \boldsymbol{c}_t; \boldsymbol{\theta}_y) \tag{18}$$

$$\boldsymbol{s}_t = f(\widetilde{\boldsymbol{y}}_{1:t-1}; \boldsymbol{\theta}_s) \tag{19}$$

$$\widetilde{\boldsymbol{y}}_t = \begin{cases} \boldsymbol{y}_t & \text{with probability} \quad \epsilon \\ \hat{\boldsymbol{y}}_t & \text{with probability} \quad 1 - \epsilon \end{cases} \tag{20}$$

$\epsilon$ gradually decays from 1 to 0 with a heuristic schedule. Considering that during training, $\widetilde{\boldsymbol{y}}_{1:t-1}$ is mostly an inconsistent mixture of the reference output and the generated output, a natural extension is sequence-level scheduled sampling (Bengio et al., 2015), where the decision is made for each sequence instead of token:

$$\widetilde{\boldsymbol{y}}_{1:t-1} = \begin{cases} \boldsymbol{y}_{1:t-1} & \text{with probability} \quad \epsilon \\ \hat{\boldsymbol{y}}_{1:t-1} & \text{with probability} \quad 1 - \epsilon \end{cases} \tag{21}$$

This type of training improves the results of many experiments, but sometimes leads to worse results (Wang et al., 2017; Bengio et al., 2015). One concern is that the decay schedule does not fit the learning pace of the model.

Professor forcing (Lamb et al., 2016) is an alternative trade-off. During training, the model $\boldsymbol{\theta}$ is viewed as a generator, which generates two output sequences for each input sequence, respectively in teacher forcing mode and free running mode[1]. For the training example $\{\boldsymbol{y}_{1:T}^{(n)}, \boldsymbol{x}_{1:L}^{(n)}\}$, let $\boldsymbol{y}_{1:T}'^{(n)}$ denote the output generated in teacher forcing mode, and $\hat{\boldsymbol{y}}_{1:T}^{(n)}$ the output generated in free running forcing mode, this can be expressed as:

$$\forall_t \; \boldsymbol{y}_t'^{(n)} \sim p(\boldsymbol{y}_t|\boldsymbol{y}_{1:t-1}^{(n)}, \boldsymbol{x}_{1:L}^{(n)}; \boldsymbol{\theta}) \tag{22}$$

$$\forall_t \; \hat{\boldsymbol{y}}_t^{(n)} \sim p(\boldsymbol{y}_t|\hat{\boldsymbol{y}}_{1:t-1}^{(n)}, \boldsymbol{x}_{1:L}^{(n)}; \boldsymbol{\theta}) \tag{23}$$

In addition to the final output, some intermediate output sequences are saved. Let $\boldsymbol{\beta}_{1:T}'^{(n)}$ and $\hat{\boldsymbol{\beta}}_{1:T}^{(n)}$ denote the intermediate output sequences generated respectively in teacher forcing and free running mode. These generated sequences form a dataset $\{\boldsymbol{y}_{1:T}'^{(n)}, \boldsymbol{\beta}_{1:T}'^{(n)}, \hat{\boldsymbol{y}}_{1:T}^{(n)}, \hat{\boldsymbol{\beta}}_{1:T}^{(n)}\}_1^N$ that is used to train a discriminator $\boldsymbol{\psi}$. $\boldsymbol{\psi}$ is trained to predict the probability that a group of sequences is generated in teacher forcing mode, and the loss function is:

$$\mathcal{L}_{\psi}(\boldsymbol{\psi}|\boldsymbol{\theta}) = -\sum_{n=1}^{N} \Big( \log\big(f(\boldsymbol{y}_{1:T}'^{(n)}, \boldsymbol{\beta}_{1:T}'^{(n)}; \boldsymbol{\psi})\big) + \log\big(1 - f(\hat{\boldsymbol{y}}_{1:T}^{(n)}, \hat{\boldsymbol{\beta}}_{1:T}^{(n)}; \boldsymbol{\psi})\big) \Big) \tag{24}$$

While this loss function is optimized w.r.t. $\boldsymbol{\psi}$, it depends on $\boldsymbol{\theta}$, hence the notation $\boldsymbol{\psi}|\boldsymbol{\theta}$. For the generator $\boldsymbol{\theta}$, there are three training objectives. The first one is the standard likelihood shown in equation 13. The second one is to fool the discriminator in free running mode:

$$\mathcal{L}_{\beta}^{(\text{F})}(\boldsymbol{\theta}|\boldsymbol{\psi}) = -\sum_{n=1}^{N} \log\big(f(\hat{\boldsymbol{y}}_{1:T}^{(n)}, \hat{\boldsymbol{\beta}}_{1:T}^{(n)}; \boldsymbol{\psi})\big) \tag{25}$$

The third one, which is optional, is to fool the discriminator in teacher forcing mode:

$$\mathcal{L}_{\beta}^{(\text{T})}(\boldsymbol{\theta}|\boldsymbol{\psi}) = -\sum_{n=1}^{N} \log\big(1 - f(\boldsymbol{y}_{1:T}'^{(n)}, \boldsymbol{\beta}_{1:T}'^{(n)}; \boldsymbol{\psi})\big) \tag{26}$$

This approach makes the distribution $p(\boldsymbol{y}_t|\hat{\boldsymbol{y}}_{1:t-1}, \boldsymbol{x}_{1:L}; \boldsymbol{\theta})$ estimated in free running mode similar to the corresponding distribution $p(\boldsymbol{y}_t|\boldsymbol{y}_{1:t-1}, \boldsymbol{x}_{1:L}; \boldsymbol{\theta})$ estimated in teacher forcing mode. In addition, it regularizes some hidden layers, encouraging them to behave as if in teacher forcing mode. The disadvantage is that it requires designing and training the discriminator.

## 3 ATTENTION FORCING

### 3.1 GUIDING THE MODEL WITH ATTENTION

For attention-based seq2seq generation, we propose a new algorithm: attention forcing. The basic idea is to use reference attention (i.e. reference alignment) and generated output to guide the model during training. In attention forcing mode, the model does not need to learn to simultaneously infer

---

[1]The term "teacher forcing", as well as "attention forcing", can refer to either an operation mode, or the approach to train a model in that operation mode. An operation mode can be used not only to train a model, but also to generate from it. For example, in teacher forcing mode, given the reference output $\boldsymbol{y}_{1:T}$, a model can generate a guided output $\boldsymbol{y}_{1:T}'$, without evaluating the loss. $\boldsymbol{y}_{1:T}'$ is likely to be different but similar to $\boldsymbol{y}_{1:T}$, and can be useful for training the discriminator.

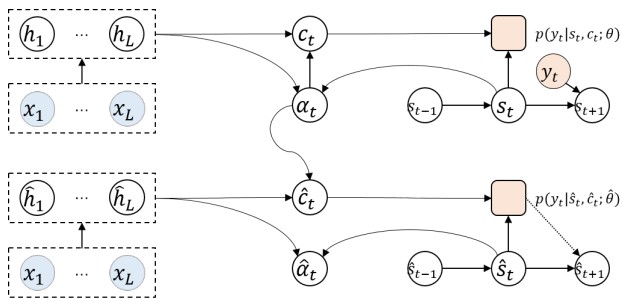

Figure 2: Illustration of attention forcing

the output and align it with the input. As the reference alignment is known, the decoder can focus on inferring the output, and the attention mechanism can focus on generating the correct alignment.

Let $\hat{\boldsymbol{\theta}}$ denote the model that is trained in attention forcing mode, and later used for inference. In attention forcing mode, $p(\boldsymbol{y}_t|\boldsymbol{y}_{1:t-1}, \boldsymbol{x}_{1:L}; \hat{\boldsymbol{\theta}})$ is estimated with the generated output $\hat{\boldsymbol{y}}_{1:t-1}$ and the reference alignment $\boldsymbol{\alpha}_t$, and equation 5 becomes:

$$p(\boldsymbol{y}_t|\boldsymbol{y}_{1:t-1}, \boldsymbol{x}_{1:L}; \hat{\boldsymbol{\theta}}) \approx p(\boldsymbol{y}_t|\hat{\boldsymbol{y}}_{1:t-1}, \boldsymbol{\alpha}_t, \boldsymbol{x}_{1:L}; \hat{\boldsymbol{\theta}}) \approx p(\boldsymbol{y}_t|\hat{\boldsymbol{s}}_t, \hat{\boldsymbol{c}}_t; \hat{\boldsymbol{\theta}}_y) \qquad (27)$$

$\hat{\boldsymbol{s}}_t$ and $\hat{\boldsymbol{c}}_t$ denote the state vector and context vector generated by $\hat{\boldsymbol{\theta}}$. Details of attention forcing can be illustrated by figure 2, as well as the following equations:

$$\boldsymbol{h}_{1:L} = f(\boldsymbol{x}_{1:L}; \boldsymbol{\theta}_h) \qquad\qquad \hat{\boldsymbol{h}}_{1:L} = f(\boldsymbol{x}_{1:L}; \hat{\boldsymbol{\theta}}_h) \qquad (28)$$

$$\boldsymbol{s}_t = f(\boldsymbol{s}_{t-1}, \boldsymbol{y}_{t-1}; \boldsymbol{\theta}_s) \qquad\qquad \hat{\boldsymbol{s}}_t = f(\hat{\boldsymbol{s}}_{t-1}, \hat{\boldsymbol{y}}_{t-1}; \hat{\boldsymbol{\theta}}_s) \qquad (29)$$

$$\boldsymbol{\alpha}_t = f(\boldsymbol{s}_t, \boldsymbol{h}_{1:L}; \boldsymbol{\theta}_\alpha) \qquad\qquad \hat{\boldsymbol{\alpha}}_t = f(\hat{\boldsymbol{s}}_t, \hat{\boldsymbol{h}}_{1:L}; \hat{\boldsymbol{\theta}}_\alpha) \qquad (30)$$

$$\hat{\boldsymbol{c}}_t = \sum_{l=1}^{L} \alpha_{t,l}\hat{\boldsymbol{h}}_l \qquad (31)$$

$$\hat{\boldsymbol{y}}_t \sim p(\boldsymbol{y}_t|\hat{\boldsymbol{s}}_t, \hat{\boldsymbol{c}}_t; \hat{\boldsymbol{\theta}}_y) \qquad (32)$$

The right side of the equations 28 to 30, as well as equations 31 and 32, show how the attention forcing model $\hat{\boldsymbol{\theta}}$ operates. $\hat{\boldsymbol{h}}_l$ and $\hat{\boldsymbol{\alpha}}_t$ denote the encoding and alignment vectors generated by $\hat{\boldsymbol{\theta}}$. $\hat{\boldsymbol{s}}_t$ is computed with $\hat{\boldsymbol{y}}_{1:t-1}$. While an alignment $\hat{\boldsymbol{\alpha}}_t$ is generated by $\hat{\boldsymbol{\theta}}$, it is not used by the decoder, because $\hat{\boldsymbol{c}}_t$ is computed with the reference alignment $\boldsymbol{\alpha}_t$. In most cases, $\boldsymbol{\alpha}_t$ is not available. One option of obtaining it is shown by the left side of equations 28 to 30, which is the same as equations 8 to 10. The option is to generate $\boldsymbol{\alpha}_t$ from a teacher forcing model $\boldsymbol{\theta}$. $\boldsymbol{\theta}$ is trained in teacher forcing mode, as described in section 2.2. Once trained, it can generate $\boldsymbol{\alpha}_t$, again in teacher forcing mode.

During inference, the attention forcing model operates in free running mode. In this case, equation 31 becomes $\hat{\boldsymbol{c}}_t = \sum_{l=1}^{L} \hat{\alpha}_{t,l}\hat{\boldsymbol{h}}_l$. The decoder is guided by $\hat{\boldsymbol{\alpha}}_t$, instead of $\boldsymbol{\alpha}_t$.

During training, there are two objectives: to infer the reference output and to imitate the reference alignment. For the first objective, the loss function is:

$$\mathcal{L}_y^{(\text{A})}(\boldsymbol{\theta}, \hat{\boldsymbol{\theta}}) = -\sum_{n=1}^{N} \sum_{t=1}^{T} \log p(\boldsymbol{y}_t^{(n)}|\hat{\boldsymbol{y}}_{1:t-1}^{(n)}, \boldsymbol{\alpha}_t^{(n)}, \boldsymbol{x}_{1:L}^{(n)}; \boldsymbol{\theta}, \hat{\boldsymbol{\theta}}) \qquad (33)$$

For the second objective, as an alignment corresponds to a categorical distribution, the loss function is the average KL-divergence between the reference alignment and the generated alignment:

$$\mathcal{L}_\alpha^{(\text{A})}(\boldsymbol{\theta}, \hat{\boldsymbol{\theta}}) = \sum_{n=1}^{N} \sum_{t=1}^{T} \text{KL}(\boldsymbol{\alpha}_t^{(n)}||\hat{\boldsymbol{\alpha}}_t^{(n)}) = \sum_{n=1}^{N} \sum_{t=1}^{T} \sum_{l=1}^{L} \alpha_{t,l}^{(n)} \log \frac{\alpha_{t,l}^{(n)}}{\hat{\alpha}_{t,l}^{(n)}} \qquad (34)$$

The two losses can be jointly optimized as $\mathcal{L}_{y,\alpha}^{(\text{A})} = \mathcal{L}_y^{(\text{A})} + \gamma\mathcal{L}_\alpha^{(\text{A})}$. $\gamma$ is a scaling factor that should be set according to the dynamic range of the two losses, which roughly indicates the norm of the gradient. The alignment loss $\mathcal{L}_\alpha^{(\text{A})}$ can be interpreted as a regularization term, which encourages the attention mechanism of $\hat{\boldsymbol{\theta}}$ to behave like that of $\boldsymbol{\theta}$. Our default optimization option is as follows. $\boldsymbol{\theta}$ is trained in teacher forcing mode, with the loss $\mathcal{L}_y^{(\text{T})}$ shown in equation 13, and then fixed to generate the reference attention. $\hat{\boldsymbol{\theta}}$ is trained with the joint loss $\mathcal{L}_{y,\alpha}^{(\text{A})}$. In our experiments, this option makes training more stable, most probably because the reference attention is the same from epoch to epoch. There are several alternative options. One example is to tie $\boldsymbol{\theta}$ and $\hat{\boldsymbol{\theta}}$, i.e. use only one set of model parameters, and train it with the joint loss $\mathcal{L}_{y,\alpha}^{(\text{A})}$. This option is less stable, but more efficient.

### 3.2 COMPARISON WITH RELATED APPROACHES

Intuitively, attention forcing, as well as scheduled sampling and professor forcing, is in the middle of teacher forcing and free running. Unlike scheduled sampling, attention forcing does not require a decay schedule, which can be difficult to tune. While the scaling factor $\gamma$ is hyper parameter, it can be set according to the dynamic ranges of the two losses, as described in section 3.1. In addition, it can be tuned according the alignment vector, which is an interpretable indicator of how well the attention mechanism works.

Beam Search Optimization (BSO) (Wiseman & Rush, 2016) is an alternative approach to dealing with the discrepancy between training and inference. The basic idea is to approximate beam search during training and penalize the reference output falling off the beam. A major difference between BSO and attention forcing is that BSO is designed for tasks where the output space is discrete, so that beam search can be used. In contrast, attention forcing is agnostic to whether the output space is continuous or discrete.

Ranzato et al. (2015) and Bahdanau et al. (2016) introduced approaches where Reinforcement Learning (RL) is adopted to deal with the discrepancy. The basic idea is to pretrain the model with teacher forcing, and the refine the model with RL. For these approaches, pretraining is essential because RL can be challenging in large action spaces, which is common for many seq2seq tasks including TTS, NMT and ASR. If the RL reward is defined as the evaluation metric at inference stage, these approaches can be considered a type of minimum Bayes risk training. Compared with these approaches, attention forcing is expected to be more stable as the training reward is naturally less sparse.

In terms of regularization, attention forcing is similar to professor forcing. The output layer of the attention mechanism, which can be viewed as a special hidden layer, is encouraged to behave as if in teacher forcing mode. The difference is that attention forcing does not require a discriminator to learn a loss function, as the KL-divergence is natural loss function for the alignment vector. The effect of regularization on the attention mechanism has been studied in previous work (Yu et al., 2017; Liu et al., 2016; Bao et al., 2018), where alternative approaches of obtaining reference attention are introduced. The approaches in Bao et al. (2018) and Yu et al. (2017) require collecting extra data for reference attention, and that in Liu et al. (2016) uses a statistical machine translation model to estimate them. In contrast, we propose to generate reference attention with a teacher forcing model, which can be trained simultaneously with the attention forcing model.

A limitation of attention forcing is that it is less general than the approaches described in section 2.2, which are well defined for all auto-regressive models, with or without attention mechanism. To apply attention forcing to a model without attention mechanism, attention needs to be defined first. For convolutional neural networks, for example, attention maps can be defined based on activation or gradient (Zagoruyko & Komodakis, 2016).

## 4 APPLICATION TO SPEECH SYNTHESIS

Attention forcing has a feature that is essential for many cascaded systems: when the reference alignment is available, the output can be generated in attention forcing mode, and will be aligned with the reference. TTS is a typical example. For TTS, the task is to map a sequence of characters $x_{1:L}$ to a sequence of waveform samples $w_{1:J}$. Directly mapping $x_{1:L}$ to $w_{1:J}$ is difficult because the two sequences are not aligned and are orders of magnitude different in length. (10 characters can correspond to more than 1000 waveform samples.) As shown in figure 3, TTS is often realized by first mapping $x_{1:L}$ to a vocoder feature sequence $y_{1:T}$, and then mapping $y_{1:T}$ to $w_{1:J}$. The vocoder feature sequence is a compact and interpretable representation of the waveform; a vocoder can be used to map vocoder features to waveform or reversely, with a series of signal processing techniques. Each feature frame corresponds to a window of waveform samples, i.e. each time step in the feature sequence corresponds to a fixed number of time steps in the waveform sequence.

The model mapping $x_{1:L}$ to $y_{1:T}$ can be referred to as the frame-level model $\theta$, and the model mapping $y_{1:T}$ to $w_{1:J}$ can be referred to as the waveform-level model $\phi$. Conventionally, $\phi$ is a vocoder, and is not learnable. $\theta$ contains a text processing frontend, a duration model and a feature model (Li et al., 2018). The text processing frontend extracts linguistic features from $x_{1:L}$; the

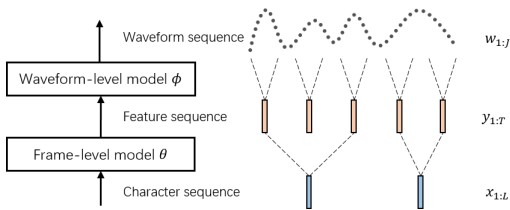

Figure 3: Illustration of a speech synthesis system

duration model predicts the duration of each linguistic feature; the feature model maps the linguistic features to $\boldsymbol{y}_{1:T}$. This paper focuses on the state-of-the-art approach, where $\boldsymbol{\theta}$, as well as $\boldsymbol{\phi}$, is a neural network. $\boldsymbol{\phi}$ can be considered a neural vocoder, which is not limited by the assumptions made by the conventional vocoders (Lorenzo-Trueba et al., 2018; Kalchbrenner et al., 2018). $\boldsymbol{\theta}$ is an attention-based seq2seq model, as described in section 2.1. Compared with the conventional approach, the attention-based model has several advantages, such as performance gain and less need for data labeling (Wang et al., 2017). Note that as shown in figure 3, $\boldsymbol{\theta}$ learns not only to map a character sequence to a feature sequence, but also to align them. In contrast, $\boldsymbol{\phi}$ does not align its input and output (Shen et al., 2018; Oord et al., 2016).

The training dataset $\{\boldsymbol{w}_{1:J}^{(n)}, \boldsymbol{x}_{1:L}^{(n)}\}_1^N$ usually contains pairs of waveform $\boldsymbol{w}_{1:J}^{(n)}$ and text $\boldsymbol{x}_{1:L}^{(n)}$. (To simplify notations, the superscript $^{(n)}$ is be omitted by default in the following discussion.) For each $\boldsymbol{w}_{1:J}$, a vocoder feature sequence $\boldsymbol{y}_{1:T}$ can be extracted. The frame-level model $\boldsymbol{\theta}$ is trained with $\{\boldsymbol{y}_{1:T}, \boldsymbol{x}_{1:L}\}$. The waveform-level model $\boldsymbol{\phi}$ can be trained with $\{\boldsymbol{w}_{1:J}, \boldsymbol{y}_{1:T}\}$, or $\{\boldsymbol{w}_{1:J}, \hat{\boldsymbol{y}}_{1:T}\}$, where $\hat{\boldsymbol{y}}_{1:T}$ is generated by $\boldsymbol{\theta}$. Training with $\hat{\boldsymbol{y}}_{1:T}$ allows $\boldsymbol{\phi}$ to fix some mistakes made by $\boldsymbol{\theta}$, but this is only possible when $\hat{\boldsymbol{y}}_{1:T}$ is aligned with $\boldsymbol{w}_{1:J}$. To ensure the alignment, the standard approach is to train $\boldsymbol{\theta}$ in teacher forcing mode, and then generate from it in the same mode. This paper proposes an alternative approach: to use attention forcing instead of teacher forcing. As analyzed in section 3.1, training $\boldsymbol{\theta}$ with attention forcing improves its performance. Furthermore, in attention forcing mode, each output $\hat{\boldsymbol{y}}_t$ is predicted based on $\hat{\boldsymbol{y}}_{1:t-1}$ (instead of $\boldsymbol{y}_{1:t-1}$), hence $\hat{\boldsymbol{y}}_{1:T}$ is more likely (than in teacher forcing mode) to contain errors that $\boldsymbol{\theta}$ makes at inference stage. Training $\boldsymbol{\phi}$ with $\hat{\boldsymbol{y}}_{1:T}$ can enable it to correct the errors, improving the quality of the waveform. Note that if $\boldsymbol{\theta}$ is trained with scheduled sampling or professor forcing, it is often not possible to predict, based only on generated output history, a vocoder feature sequence aligned with the reference waveform. Also note that $\boldsymbol{\phi}$ is trained in teacher forcing mode, as it does not have attention mechanism. Hence the rest of this section focuses on discussing $\boldsymbol{\theta}$ at training stage and inference stage.

During training, it is often assumed that the output tokens follow a certain type of distribution, so that minimizing the loss $\mathcal{L}_y^{(\mathtt{A})}$ shown in equation 33 can be approximated by minimizing some distance metric between $\boldsymbol{y}_{1:T}$ and $\hat{\boldsymbol{y}}_{1:T}$. For example, assuming that the distribution shown in equation 27 is a Laplace distribution, minimizing $\mathcal{L}_y^{(\mathtt{A})}$ is equivalent to minimizing the average $\ell_1$ distance:

$$\operatorname*{argmin}_{\boldsymbol{\theta},\hat{\boldsymbol{\theta}}} \mathcal{L}_y^{(\mathtt{A})}(\boldsymbol{\theta}, \hat{\boldsymbol{\theta}}) \approx \operatorname*{argmin}_{\boldsymbol{\theta},\hat{\boldsymbol{\theta}}} \sum_{n=1}^N \sum_{t=1}^T ||\boldsymbol{y}_t^{(n)} - \hat{\boldsymbol{y}}_t^{(n)}||_1 \tag{35}$$

$$\hat{\boldsymbol{y}}_t = \operatorname*{argmax}_{\boldsymbol{y}_t} p(\boldsymbol{y}_t|\hat{\boldsymbol{y}}_{1:t-1}, \boldsymbol{\alpha}_t, \boldsymbol{x}_{1:L}; \hat{\boldsymbol{\theta}}) \tag{36}$$

The notation is the same as in section 3.1. $\hat{\boldsymbol{\theta}}$ denotes the attention forcing model; $\boldsymbol{\theta}$ denotes the teacher forcing model generating reference alignment. Equation 36 replaces equation 32. In this case, $\hat{\boldsymbol{y}}_t$ is not sampled, and is always the mode of the predicted distribution. During inference, the exact search (equation 4) is approximated by greedy search: (Note that for TTS, the main difference between training and inference is the alignment, which influences duration more than quality.)

$$\forall_t \, \hat{\boldsymbol{y}}_t = \operatorname*{argmax}_{\boldsymbol{y}_t} p(\boldsymbol{y}_t|\hat{\boldsymbol{y}}_{1:t-1}, \hat{\boldsymbol{\alpha}}_t, \boldsymbol{x}_{1:L}^*; \hat{\boldsymbol{\theta}}) \tag{37}$$

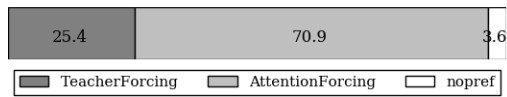

Figure 4: Result of the listening test comparing teacher forcing and attention forcing

## 5 EXPERIMENTS

### 5.1 SPEECH SYNTHESIS

The TTS experiments are conducted on LJ dataset (Ito, 2017), which contains 13,100 utterances from a single speaker. The utterances vary in length from 1 to 10 seconds, totaling approximately 24 hours. A transcription is provided for each waveform, and the corresponding vocoder features are extracted with PML vocoder (Degottex et al., 2016). The training-validation-test split is 13000-50-50. The waveform-level model is the Hierarchical Recurrent Neural Network (HRNN) neural vocoder (Mehri et al., 2016; Dou et al., 2018). The model structure is exactly the same as described in Dou et al. (2018), and the model configuration is adjusted for efficiency. The frame-level model is very similar to Tacotron (Wang et al., 2017). The model structure and configuration are the same as described in Wang et al. (2017), except that: 1) the decoder target is vocoder features; 2) the attention mechanism is the hybrid (content-based + location-based) attention (Chorowski et al., 2015); 3) each decoding step predicts 5 vocoder feature frames. The neural vocoder is always trained with teacher forcing. The frame-level model is trained with either teacher forcing or attention forcing. Details of the setup (data, models and training) are presented in appendix A.2.1.

Two TTS systems are built: a teacher forcing system and an attention forcing system. For the teacher forcing system, the frame-level model $\boldsymbol{\theta}$ is trained in teacher forcing mode. The neural vocoder $\boldsymbol{\phi}$ is trained with the vocoder features generated (in teacher forcing mode) by $\boldsymbol{\theta}$. For the attention forcing system, the frame-level model $\hat{\boldsymbol{\theta}}$ is trained in attention forcing mode, with reference attention generated (in teacher forcing mode) by $\boldsymbol{\theta}$. At this stage, $\hat{\boldsymbol{\theta}}$ is updated, while $\boldsymbol{\theta}$ is fixed. The neural vocoder $\hat{\boldsymbol{\phi}}$ is trained with the vocoder features generated (in attention forcing mode) by $\hat{\boldsymbol{\theta}}$. At inference stage, all the models operate in free-running mode.

For TTS, human perception is the gold-standard. The two systems are compared in a subjective listening test. Over 30 workers from Amazon Mechanical Turk are instructed to listen to pairs of utterances, and indicate which one they prefer in terms of overall quality. Each comparison includes 5 pairs of utterances randomly selected among all the test utterances. Figure 4 shows the result of the listening test. Each number indicates the percentage of a certain preference. Most participants prefer attention forcing. We strongly encourage readers to listens to the generated utterances[2]. It is obvious that attention forcing yields utterances that are significantly more natural and expressive.

### 5.2 MACHINE TRANSLATION

The NMT experiments are conducted on the English-to-Vietnamese task in IWSLT 2015. It is a low resource NMT task, where training set contains 133K sentence pairs. The Stanford pre-processed data is used. The TED tst2012 is used as a validation set, and BLEU scores on TED tst2013 are reported. The scores use a 4-gram corpus level BLEU with equal weights. Google's attention-based encoder-decoder LSTM model (Wu et al., 2016) is adopted. Details of the setup (data, model and training) are presented in appendix A.2.2.

Our initial experiments show that directly applying attention forcing to NMT can degrade the performance. One concern is that for translation, various re-orderings of the output sequence are valid. In this case, guiding the model with generated output can be problematic, as the reference output can take an ordering that is different from the generated output. To see if this is the reason, we tried a modified attention forcing mode, where the model is guided with reference attention and reference output. The right side of equation 29 becomes: $\hat{\boldsymbol{s}}_t = f(\hat{\boldsymbol{s}}_{t-1}, \boldsymbol{y}_{t-1}; \hat{\boldsymbol{\theta}}_s)$. $\hat{\boldsymbol{s}}_t$ is computed with

---

[2]Generated test utterances are randomly selected and made available at `http://mi.eng.cam.ac.uk/~qd212/iclr2020/samples.html`

the reference output $\boldsymbol{y}_{1:t-1}$, and matches the reference attention $\boldsymbol{\alpha}_t$ Other parts of attention forcing (equations 28 to 31) stay the same, hence $\hat{\boldsymbol{y}}_t$ is predicted with $\boldsymbol{y}_{1:t-1}$ and $\boldsymbol{\alpha}_t$.

In the following experiments, two NMT models are compared: one is trained in teacher forcing mode, with the NLL loss in equation 13; the other is trained in the modified attention forcing mode described above, with both the NLL loss and the attention loss in equation 34. An ensemble of 10 models are trained with teacher forcing. Then each model generates reference attention for a corresponding model trained with additional attention loss. The average performance of the teacher forcing models is 26.35 BLEU, and adding the attention loss yields an average +0.35 BLEU gain. 9 of out 10 times, the performance improves. The slight but consistent gain shows that for NMT, guiding the model with generated output is indeed the cause degrading the performance. It also shows that guiding the model with reference attention can be beneficial. One possible reason is that the attention loss regularizes the attention mechanism. Another is that the model does not need to learn to simultaneously infer the output and align it with the input.

## 6 CONCLUSION

This paper introduces attention forcing, which guides a seq2seq model with generated output history and reference attention. This approach can train the model to recover from its mistakes, in a stable fashion, without the need for a schedule or a classifier. In addition, it allows the model to generate output sequences aligned with the reference output sequences, which can be important for cascaded systems like many TTS systems. The TTS experiments show that attention forcing yields significant gain in speech quality. The NMT experiments show that for tasks where various re-orderings of the output are valid, guiding the model with generated output history can be problematic, while guiding the model with reference attention yields slight but consistent gain in BLEU score.

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

# A APPENDIX

## A.1 DETAILS OF SEQUENCE-TO-SEQUENCE GENERATION

The exact search shown in equation 4 is computationally expensive, and is often approximated by greedy search if the output space is continuous, or beam search if the output space is discrete (Bengio et al., 2015). For greedy search, the model generates the output sequence one token at a time based on previous output tokens, until a special end-of-sequence token is generated. For beam search, a heap of $b$ best candidate sequences is kept. At each time step, the candidates are updated by extending each candidate by one step, and pruning the heap to only keep $b$ best candidates. The beam search stops when no new sequences are added.

## A.2 DETAILS OF EXPERIMENTAL SETUP

### A.2.1 SPEECH SYNTHESIS

The TTS experiments are conducted on LJ dataset (Ito, 2017). This public domain dataset contains 13,100 utterances from a single speaker reading passages from 7 non-fiction books. The utterances vary in length from 1 to 10 seconds and have a total length of approximately 24 hours. A transcription (character sequence) is provided for each utterance (waveform sequence). The waveforms are resampled to 16kHz to increase the efficiency of neural vocoders. Corresponding vocoder features are extracted at the frame rate of 0.2kHz, using a PML vocoder (Degottex et al., 2016). The training-validation-test split is 13000-50-50.

The frame-level model is very similar to Tacotron (Wang et al., 2017), a powerful attention-based encoder-decoder model. The differences are: 1) the decoder target is vocoder features; 2) the attention mechanism is the hybrid (content-based + location-based) attention described in Chorowski et al. (2015); 3) the reduction factor is 5, i.e. each decoding step predicts 5 vocoder feature frames. Apart from these, the model structure is the same as described in Wang et al. (2017). The input characters are represented as one-hot vectors. The encoder has an embedding layer mapping the one-hot vectors to continuous vectors, a bottleneck layer with dropout, and a CBHG module generating the final encoding sequence. The CBHG module consists of a bank of 1-D convolutional filters, followed by highway networks (Srivastava et al., 2015) and a bidirectional GRU. The decoder has a stack of GRUs with vertical residual connections and generates the intermediate vocoder features. These features are post-processed by another CBHG module, yielding the final vocoder features. The model configuration is the same as described by Table 1 in Wang et al. (2017).

The waveform-level model is the Hierarchical Recurrent Neural Network (HRNN) neural vocoder (Mehri et al., 2016; Dou et al., 2018). The HRNN structure is a hierarchy of tiers; each tier includes several neural network layers and operates at a different frequency. The lowest tier operates at waveform-level frequency, and outputs distributions of waveform samples. Each higher tier operates at a lower frequency, and supervises the tier below it. The model configuration is as follows. Tier 0 is a 4-layer DNN, including three fully connected layers with ReLU activation and a softmax output layer; the dimension is 1024 for the first two fully connected layers, and is 256 for the other two

layers. The other tiers are all 1-layer RNNs; Gated Recurrent Unit (GRU) (Chung et al., 2014) is used and the dimension is 512 for all layers. The frequencies for tiers 0 to 3 are respectively 16kHz, 8kHz, 2kHz and 0.4kHz. This neural vocoder models each waveform sample with a categorical distribution. Hence the waveform samples are quantized into 256 integer values.

The frame-level model is trained with either teacher forcing or attention forcing. In both cases, the $\ell_1$ loss shown in equation 35 is used for both the decoder and post-processing CBHG. The two losses have equal weights. For attention forcing, the additional alignment loss shown in equation 34 is used for the attention mechanism, and the scaling factor $\gamma$ is 50. The neural vocoder is always trained in teacher forcing mode, and the loss function is shown in equation 13. For all experiments, the optimizer is Adam (Kingma & Ba, 2014), and the initial learning rate is 0.001.

### A.2.2 MACHINE TRANSLATION

The NMT experiments are conducted on the English-to-Vietnamese task in IWSLT 2015. It is a low resource NMT task, with the parallel training set containing 133K sentence pairs. The Stanford pre-processed data (`https://nlp.stanford.edu/projects/nmt/`) is used. The attention-based encoder-decoder LSTM model (Wu et al., 2016) is adopted. The model is simplified with a smaller number of LSTM layers due to the small scale of data: the encoder has 2 layers of bi-LSTM and the decoder has 4 layers of uni-LSTM; the general form of Luong attention Luong et al. (2015) is used; both English and Vietnamese word embeddings have 200 dimensions and are randomly initialised. Adam optimiser is used with a learning rate of 0.002 and the maximum gradient norm is set to be 1. Dropout is used with a probability of 0.2. During inference, predictions are made using beam search with a width of 10.

