# OpenReview forum: "Attention Forcing for Sequence-to-sequence Model Training"
_ICLR.cc/2020/Conference — Reject_

### Official Review · AnonReviewer3 · 2019-10-20
**Official Blind Review #3**

**Rating:** 1

**Review:**

This paper proposes an alternative mechanism of training the attention values of a sequence to sequence learning model as applied to tasks like speech synthesis and translation.  During training they compute two forms of attention: (1) the standard soft-attention from a decoder fed with teacher forced output, and (2) the inference-time attention from a decoder fed with predicted outputs.  Their training objective consists of two terms: The first is the token-wise cross entropy loss but by conditioning on the predicted output  but with teacher-forced attention.  The second is a KL distance between the above two types of attention distributions.   Experiments with mechanical  turks indicate that their attention forcing mechanism is strongly preferred over the existing teacher forced output and attention model.  On translation their method provides little or no improvement.

I am inclined towards rejecting the paper because the experiment and related work section still requires a lot of work before 1. The claimed utility of the idea is established, and 2. The novelty over the many existing attention architectures is established.   I elaborate on each of these next.

Related work: Recently, many papers have directly or indirectly handled the problem of exposure bias that this paper attempts to address.  The paper does not discuss most of these.  Here are some that are missed from the paper:

1.   Sequence level training with recurrent neural networks
MA Ranzato, S Chopra, M Auli, W Zaremba, 2015.
This paper shows that the scheduled sampling method (discussed in the paper) is much worse than a reinforce-based training mechanism of handling exposure bias.

2. An actor-critic algorithm for sequence prediction
D Bahdanau, P Brakel, K Xu, A Goyal, R Lowe

3.  Posterior Attention Models for Sequence to Sequence Learning
S Shankar, S Sarawagi - 2019

4. Latent Alignment and Variational Attention
Yuntian Deng, Yoon Kim, Justin Chiu, Demi Guo, Alexander M. Rush 2018

5. Attention Focusing for Neural Machine Translation by Bridging Source and Target Embeddings
Shaohui Kuang, Junhui Li, António Branco, Weihua Luo, Deyi Xiong

Experiments:  Their experiments are rather sketchy and limited.
The TTS experiments are only on one dataset.  Their method is compared only with the standard seq2seq learning approach.  Even the scheduled sampling or professor forcing methods are not compared with.  In addition, state of the art TTS methods have gained significantly from hierarchical attention.  As such as far as the TTS task is concerned the significance of the improved quality over a baseline seq2seq method is limited.

For translation they consider only the English-Vietnamese task whereas there are tens of other translation tasks that are used in recent literature.

Overall, the idea proposed seems quite incremental, experiments are limited, and related work discussion incomplete.

*********
I read the author response but I do not think the paper is ready for publication yet without the thorough comparison with related work.


**Experience Assessment:**

I have published in this field for several years.

**Review Assessment: Checking Correctness Of Derivations And Theory:**

I assessed the sensibility of the derivations and theory.

**Review Assessment: Checking Correctness Of Experiments:**

I assessed the sensibility of the experiments.

**Review Assessment: Thoroughness In Paper Reading:**

I read the paper at least twice and used my best judgement in assessing the paper.

---

> ### Author Response · Authors · 2019-11-15
> **We thank the reviewer for the insightful review. We address the comments and questions below.**
>
> The primary goal of attention forcing is to fixing exposure bias in seq2seq learning. [1,2] introduce alternative approaches based on Reinforcement Learning (RL). These RL-based approaches can be considered a type of minimum Bayes risk training, which is briefly discussed in section 2.2 of our paper. Compared with these approaches, attention forcing is expected to be more stable as the training reward is less sparse, and the model is trained in a mode between teacher forcing and free running. In addition,  attention forcing is more suitable for tasks where it is important for the predicted sequence to be well aligned with the reference sequence, such as TTS. [3,4,5] introduce methods to improve the attention mechanism by changing the model architecture. In contrast, attention forcing is a training method, and can be used in combination with the models introduced in [3,4,5].
>
> In our preliminary experiments, we tried to use scheduled sampling with a linear decay schedule for both TTS and NMT, which did not result in noticeable performance gain. It is however not fair to compare scheduled sampling and attention forcing based on these results, as the former requires more hyper parameter tuning. The main reason why we did not compare with scheduled sampling or professor forcing in TTS is that most, if not all, state-of-the art TTS models are trained with teacher forcing. The seq2seq model used in our work is largely based on Tacotron1&2, which are to our knowledge among the best-performing models in TTS.
>
> [1] Sequence level training with recurrent neural networks; MA Ranzato, S Chopra, M Auli, W Zaremba, 2015
> [2] An actor-critic algorithm for sequence prediction; D Bahdanau, P Brakel, K Xu, A Goyal, R Lowe
> [3] Posterior Attention Models for Sequence to Sequence Learning; S Shankar, S Sarawagi - 2019
> [4] Latent Alignment and Variational Attention; Yuntian Deng, Yoon Kim, Justin Chiu, Demi Guo, Alexander M. Rush 2018
> [5] Attention Focusing for Neural Machine Translation by Bridging Source and Target Embeddings; Shaohui Kuang, Junhui Li, António Branco, Weihua Luo, Deyi Xiong

---

### Official Review · AnonReviewer1 · 2019-10-23
**Official Blind Review #1**

**Rating:** 3

**Review:**

This paper proposes a method for fixing exposure bias (ie. training vs generated distribution mismatch) in seq2seq modeling with attention, particularly for the application of speech synthesis where reference alignments are available.


Related Work is missing:

- Another paper that studies fixing exposure bias in seq2seq learning:

Wiseman & Rush. Sequence-to-Sequence Learning as Beam-Search Optimization
https://arxiv.org/pdf/1606.02960.pdf

- Other papers that try to enforce attention to attend to specific locations:

Bao et al. Deriving Machine Attention from Human Rationales. https://arxiv.org/abs/1808.09367

Liu et al. Neural Machine Translation with Supervised Attention. https://www.aclweb.org/anthology/C16-1291/

Yu et al. Supervising Neural Attention Models for Video Captioning by Human Gaze Data. https://arxiv.org/abs/1707.06029

Without the comparison against other related papers that also aim to supervise attention mechanisms (there are other beyond the ones I cited above)s, it is unclear how much is novel about this paper.

- Furthermore, it is conceptually clear to me that attention-forcing fully matches the training vs generated distributions. The authors should describe in greater detail why this happens this, or whether these distributions are not required to fully match in attention-forcing (and in this case, why this would be desirable).

- The experiments are not very convincing (only 30 human evaluators for Speech synthesis with no other quantitative evaluation, NMT results that are not particularly promising).

- Use of non-anonymous github link is questionable for blinded submissions.

**Experience Assessment:**

I have published one or two papers in this area.

**Review Assessment: Checking Correctness Of Derivations And Theory:**

N/A

**Review Assessment: Checking Correctness Of Experiments:**

I assessed the sensibility of the experiments.

**Review Assessment: Thoroughness In Paper Reading:**

N/A

---

> ### Author Response · Authors · 2019-11-15
> **We thank the reviewer for the insightful review. We address the comments and questions below.**
>
> The primary goal of attention forcing is to fixing exposure bias in seq2seq learning. [1] introduces an alternative approach. The basic idea is to approximate beam search during training and penalize the reference output falling off the beam. A major difference between this approach and our work is that this approach is designed for tasks where the output space is discrete, so that beam search can be used. In contrast, our approach is agnostic to whether the output space is continuous or discrete. In terms of regularizing the attention mechanism, [2,3,4] are similar to our work. Regularizing the attention mechanism can be considered a special case of hidden layer regularization, which is involved in professor forcing. In the context of attention forcing, these approaches can be considered alternative ways of obtaining reference attention. We propose to generate reference attention with a teacher forcing model, which can be trained simultaneously with the attention forcing model. [2,4] requires collecting reference attention maps, and [3] uses an SMT model to estimate them.
>
> For attention forcing, there is still discrepancy between training and inference, because the reference attention is not available at inference stage. It would be desirable to eliminate this discrepancy (e.g. by training the model in free running mode), but this is likely to make training less stable.
>
> For the TTS experiment, the test set contains 50 sentences. Although there were only 36 human evaluators, each of them listened to 5 of the test sentences. So in total 150 sentences are randomly drawn from the test set and evaluated. On average, each test sentence is evaluated 3 times.
>
> [1] Wiseman & Rush. Sequence-to-Sequence Learning as Beam-Search Optimization
> [2] Bao et al. Deriving Machine Attention from Human Rationales
> [3] Liu et al. Neural Machine Translation with Supervised Attention
> [4] Yu et al. Supervising Neural Attention Models for Video Captioning by Human Gaze Data

---

### Official Review · AnonReviewer2 · 2019-10-24
**Official Blind Review #2**

**Rating:** 3

**Review:**

This paper proposes a novel training scheme for seq2seq models where attention or reference alignment is used in combination with free-running mode for improving training.

The positives of this paper are that it is well written and very clear. It also is very relevant as seq2seq models can be hard to train and techniques like scheduled sampling and x-forcing algorithms are good heuristics but heuristics none-the-less.

The downside of this paper is in the experimental results and also complexity. It would’ve been good to see a broader set of experiments to really benchmark attention-forcing from other self-attention models.

Attention forcing also requires a reference or ground-truth alignment, which is often not available. Hence the authors propose to simultaneously train another teacher-forcing model to estimate the reference alignment. However, this would incur twice the computation complexity.

Attention forcing could also be used in conjunction with scheduled sampling. How does that compare with the reported results for attention forcing?

**Experience Assessment:**

I have read many papers in this area.

**Review Assessment: Checking Correctness Of Derivations And Theory:**

I carefully checked the derivations and theory.

**Review Assessment: Checking Correctness Of Experiments:**

I assessed the sensibility of the experiments.

**Review Assessment: Thoroughness In Paper Reading:**

I read the paper at least twice and used my best judgement in assessing the paper.

---

> ### Author Response · Authors · 2019-11-15
> **We thank the reviewer for the insightful review. We address the comments and questions below.**
>
> It is true that when simultaneously training a teacher-forcing model and an attention-forcing model, each forward-backward pass requires approximately twice the computation required when training a single model. However, it should be noted that the time required remains the same, as the two models can be trained in parallel. In addition, if memory is an issue, the two models can be trained in a sequential fashion.
>
> There are multiple ways of combining scheduled sampling with attention forcing. As attention forcing is in the middle of teacher forcing and free running, a schedule can be used to shift the training scheme gradually from teacher forcing to attention forcing, and then from attention forcing to free running. We hypothesize that using a schedule to gradually shift the training scheme would make the training more stable. In particular, for tasks (e.g. NMT) where the output space is discrete and the predicted output can be categorically wrong, the gradual shift can be essential. In contrast, it is expected to be less important for tasks (e.g. TTS) where the output space is continuous and errors in the predicted output are less serious.

---

### Public Comment · ~Murali_Karthick_Baskar1 · 2019-10-01
**Kind suggestion wrt paper uploading**

Dear authors,
I am able to find your paper in arXiv and the github code link. Please remember that this removes the idea of double-blind review.

---

> ### Author Response · Authors · 2019-10-02
> **Thank you and we will make the github repository private if necessary**
>
> Dear Murali, Thank you for your comment. To the best of our knowledge, submission of the paper to archival repositories such as arXiv are allowed (according to https://iclr.cc/Conferences/2020/CallForPapers). We thought that the github code does not reveal the authors, but we might have missed some information. If necessary, we will make the repository private (invisible to the public) until the decisions are made.

---

> > ### Public Comment · ~Cantona_ViVian1 · 2019-10-02
> > **Header**
> >
> > Please change the header of your arxiv paper to under review or something else. For now, it is not a published paper in ICLR 2020.

---

> > > ### Author Response · Authors · 2019-10-02
> > > **Thank you and we will change the header**
> > >
> > > Hello Cantona, Thank you for the reminder. We have submitted a replacement, where the header is changed. The replacement is scheduled to be announced at Fri, 4 Oct 2019 00:00:00 GMT.

---

### Author Response · Authors · 2019-10-25
**Correction of typographical errors**

In the bottom right side of Figure 2, $\hat{s}_{1}$ should be $\hat{s}_{t}$.
Similarly, in the right side of Equation 30, $\hat{s}_{1:t-1}$ should be $\hat{s}_{t}$.
Appologies for these errors.

---

### Decision · Program_Chairs · 2019-12-19

**Decision:**

Reject

**Comment:**

The paper proposed an attention-forcing algorithm that guides the sequence-to-sequence model training to make it more stable. But as pointed out by the reviewers, the proposed method requires alignment which is normally unavailable. The solution to address that is using another teacher-forcing model, which can be expensive.

The major concern about this paper is the experimental justification is not sufficient:
* lack of evaluations of the proposed method on different tasks;
* lack of experiments on understanding how it interact with existing techniques such as scheduled sampling etc;
* lack of comparisons to related existing supervised attention mechanisms.